# Optimizing Nutritional Balance: Integrating the Mediterranean Diet into Low-FODMAP Nutrition

**DOI:** 10.3390/microorganisms13092085

**Published:** 2025-09-07

**Authors:** Athanasia Dimitriou, Ioanna Aggeletopoulou, Christos Triantos

**Affiliations:** Division of Gastroenterology, Department of Internal Medicine, University of Patras, 26504 Patras, Greece; diminancy@yahoo.gr (A.D.); chtriantos@upatras.gr (C.T.)

**Keywords:** irritable bowel syndrome, Mediterranean diet, low-FODMAP diet, IBS management, nutritional implications, gut microbiota

## Abstract

Irritable Bowel Syndrome (IBS) is a functional disorder of the digestive system. Its global prevalence varies widely, estimated at up to 10%, due to differences in diagnostic criteria, cultural factors, and dietary patterns. Dietary interventions have emerged as first-line strategies for symptom management, with the low-FODMAP (fermentable oligo-, di-, and monosaccharide and polyol) diet demonstrating robust efficacy in reducing gastrointestinal symptoms by minimizing poorly absorbed, fermentable carbohydrates. However, concerns regarding the long-term nutritional adequacy, potential alterations in gut microbiota composition, and sustainability of the low-FODMAP diet have prompted the need for more integrative and nutritionally balanced dietary models. In contrast, the Mediterranean Diet (MD), rich in fruits, vegetables, legumes, whole grains, nuts, olive oil, and omega-3 fatty acids from fish, is widely recognized for its anti-inflammatory, cardiometabolic, and potential anticarcinogenic effects. Notably, adherence to the MD has been associated with favorable gut microbial profiles and reduced risk of colorectal and other gastrointestinal cancers. However, the high FODMAP content of many MD components limits its immediate compatibility with IBS dietary protocols. This review explores the evidence supporting the integration of MD principles into a low-FODMAP framework for the dietary management of IBS. Rather than proposing a new model, we synthesize existing literature, including recent clinical findings, and offer practical, evidence-informed guidance for tailoring a Mediterranean-style low-FODMAP diet that supports both symptom management and long-term nutritional health. Integrating MD principles into the low-FODMAP approach offers a promising strategy to enhance the nutritional quality, sustainability, and long-term efficacy of dietary management in IBS.

## 1. Introduction

Irritable Bowel Syndrome (IBS) is a chronic functional disorder of the gastrointestinal tract characterized by symptoms such as abdominal pain, bloating, diarrhea and/or constipation, without an organic cause [1]. The Rome Foundation’s Global Epidemiological Study, utilizing the Rome IV criteria across 33 countries, revealed that prevalence rates were similar between Europe (3.5–5.9%) and the United States (3.5–5.3%), while slightly lower rates were observed in Asia (1.3–4.7%) and Australia (3.5%) [2]. IBS is often associated with anxiety, microbiome disturbance and intolerance to certain foods [1,3]. Despite its benign nature, IBS significantly affects the quality of life and daily functioning of patients.

IBS is a common condition. Its underlying mechanisms are not yet fully understood and appear to be complex and varied [4,5]. A range of factors are believed to contribute to its development, including disrupted gastrointestinal motility, increased sensitivity of the gut (visceral hypersensitivity), small intestinal bacterial overgrowth (SIBO), environmental influences, dietary patterns, food intolerances, and imbalances in the gut microbiota (dysbiosis) [6,7]. Substantial evidence points to the gut microbiota as a key player in IBS, with both compositional and functional changes being documented [8,9]. These microbial imbalances are known to exacerbate IBS symptoms, such as abdominal pain, mild inflammation, and irregular bowel movements [10].

Diet is recognized as one of the main modifiable factors associated with the intensity and frequency of symptoms. The low-FODMAP diet has emerged as the most effective approach for relieving symptoms in over 70% of patients [11]. Nevertheless, concerns are raised regarding its nutritional adequacy and long-term sustainability. In contrast, the Mediterranean Diet (MD) is a dietary pattern rich in antioxidants, fiber, good fats, and anti-inflammatory components, with recognized benefits for cardiometabolic and digestive health [12]. However, many of its staple foods are high in FODMAPs, making its implementation difficult for people with IBS.

The aim of this review is to critically examine the feasibility, clinical functionality, and nutritional adequacy of integrating principles of the MD into a structured low-FODMAP approach for individuals with IBS. This includes evaluating the potential of such a combined dietary model to preserve gastrointestinal symptom relief, while addressing known limitations of the low-FODMAP diet, such as long-term nutritional deficiencies, alterations in gut microbiota composition, and sustainability concerns. Moreover, the study aims to formulate practical, culturally relevant dietary guidelines that merge anti-inflammatory and gut-supportive elements of the Mediterranean Diet with the symptom management benefits of the low-FODMAP strategy, thereby proposing a comprehensive, patient-centered framework for nutritional management in IBS.

## 2. Irritable Bowel Syndrome (IBS)

According to the Rome IV criteria (2016), IBS is characterized by recurrent abdominal pain occurring, on average, at least one day per week in the last three months and is associated with at least two of the following features: pain related to defecation, a change in stool frequency, or a change in stool form. Those diagnostic criteria should be met for the previous three months, with symptom onset occurring at least six months prior to diagnosis [6,7].

IBS is classified based on stool consistency using the Bristol Stool Form Scale [8]. The main subtypes include, IBS with predominant constipation (IBS-C), where more than 25% of bowel movements are hard or lumpy (types 1 or 2) and less than 25% are loose or watery (types 6 or 7), IBS with predominant diarrhea (IBS-D), where more than 25% of stools are loose or watery and less than 25% are hard and mixed-type IBS (IBS-M), where more than 25% of bowel movements fall into both the hard and loose categories. A fourth subtype, IBS unclassified (IBS-U), is used when stool patterns do not meet the criteria for the other categories, typically involving less than 25% hard and less than 25% loose stools [7].

### The Gut–Brain Axis and Psychosocial Triggers in IBS

The gut–brain axis (GBA) constitutes a dynamic, bidirectional communication network linking the gastrointestinal (GI) tract with the central nervous system (CNS), integrating neural, endocrine, immune, and microbial signals. It plays a central role in monitoring GI function while simultaneously coordinating emotional and cognitive processes with intestinal activity [13]. A growing body of evidence suggests that dysregulation of the GBA is a key contributor to the pathophysiology of IBS, particularly through disruptions in gut microbiota composition and altered neuroimmune and neuroendocrine signaling [14,15]. Psychosocial stressors, including anxiety, depression, and early life adversity, can significantly affect the GBA, influencing intestinal permeability, immune activation, enteric reflexes, and microbial homeostasis [15]. Clinical and preclinical studies have shown that alterations in gut microbiota may influence CNS function and behavior through microbial metabolites, immune pathways, and vagal nerve signaling [16,17]. Moreover, up to 60% of IBS patients present with comorbid psychiatric conditions, with anxiety and depression being the most prevalent, suggesting a strong psychosomatic component to symptom severity and treatment resistance [18]. This close interplay between psychological distress, microbial dysbiosis, and gut dysfunction highlights the importance of considering the GBA as both a diagnostic framework and a therapeutic target in IBS [19].

## 3. Physiological Impact of FODMAPs in the Gastrointestinal Tract

FODMAPs include lactose, fructose (in excess), fructans (fructo-oligosaccharides), galacto-oligosaccharides (GOS), and polyols (sorbitol, mannitol, maltitol, xylitol, polydextrose, and isomalt) [20]. They are groups of short-chain carbohydrates that are poorly absorbed in the small intestine and subsequently fermented by colon bacteria, leading to the production of gas and other byproducts. This process can cause symptoms such as bloating, gas, diarrhea or constipation, especially in people with IBS or other functional gastrointestinal disorders. FODMAPs are found in commonly consumed foods including fruits, vegetables, bread, cereals, grains, nuts, seeds, dairy products, processed foods and beverages [21]. The typical daily consumption of FODMAPs in a regular diet varies between 15 and 30 g [22,23,24], though this amount can vary depending on dietary patterns and cultural differences in food consumption.

The adverse gastrointestinal effects of FODMAPs are primarily attributed to three interconnected mechanisms [25]. First, due to their molecular structure and the absence of specific transporters or enzymes, FODMAPs are incompletely absorbed in the small intestine [26]. As a result, they accumulate in the intestinal lumen, where they exert osmotic effects. This osmotic activity draws water into the lumen, increasing stool volume and accelerating intestinal transit, which can lead to diarrhea in susceptible individuals. Furthermore, once FODMAPs reach the colon, they undergo rapid fermentation by resident colonic bacteria, producing gases such as hydrogen and methane. This gas production contributes to luminal distension and triggers symptoms including bloating, abdominal discomfort, and flatulence, particularly in individuals with visceral hypersensitivity, such as those diagnosed with IBS [14]. Moreover, methane production has been associated with slowed intestinal transit and may contribute to symptoms in IBS-C, which is prevalent in Western populations. Thus, symptom manifestation likely depends on the complex interplay between FODMAP exposure, microbiota composition, gut motility, and host sensitivity [27]. Collectively, these pathophysiological mechanisms explain the symptomatic burden associated with FODMAP intake in functional gastrointestinal disorders and support the clinical basis behind dietary interventions like the low-FODMAP diet [25].

### 3.1. Lactose

Lactose is a disaccharide composed of two simple sugars, glucose and galactose. It is primarily found in dairy products such as cow’s milk, soft cheese, custard, sweetened condensed milk, evaporated milk, and ice cream [28]. In individuals who lack sufficient activity of the enzyme lactase, lactose cannot be hydrolyzed into its monosaccharide components. As a result, unabsorbed lactose remains in the intestinal lumen, where it exerts osmotic effects by drawing water into the small intestine and eventually reaching the colon, where it undergoes fermentation by gut bacteria. For individuals with lactose intolerance, this process can trigger symptoms such as bloating, gas, abdominal pain, excessive flatulence, and diarrhea [29].

Lactase deficiency may be attributed to several underlying causes [30]. The most common form is primary lactose intolerance, also referred to as adult-type hypolactasia, which occurs due to a genetically programmed decline in lactase production after weaning [30]. In rare cases, congenital lactase deficiency (CLD), an autosomal recessive disorder, results in a complete absence of lactase from birth [30]. Additionally, a secondary or temporary lactase deficiency may arise as a result of acute intestinal infections, such as viral gastroenteritis, or from chronic conditions like untreated coeliac disease or inflammatory bowel disease, where damage to the intestinal mucosa impairs lactase expression [30]. Understanding the etiology of lactase deficiency is critical for accurate diagnosis and for tailoring dietary strategies, including lactose restriction or enzyme supplementation, to alleviate symptoms and maintain nutritional adequacy.

### 3.2. Fructose

Fructose is a monosaccharide that occurs naturally in many different foods [31]. Foods high in excess fructose include apples, pears, mangoes, dried fruits, asparagus, sugar snap peas, honey, high-fructose corn syrup, and fruit juices [32]. Fructose absorption in the small intestine is facilitated by specific transporters, via the GLUT5 (high-affinity) or GLUT2 (low-affinity) transporter [33]. It is influenced by both the quantity ingested and the ratio of fructose to glucose present in the intestinal lumen. Glucose helps with the absorption of fructose through the GLUT2 transporter, so foods with a balanced 1:1 ratio of fructose to glucose are more easily tolerated. In contrast, when fructose is present in greater amounts than glucose (referred to as excess fructose, meaning more free fructose than free glucose), it is poorly absorbed, leading to increased water movement into the intestinal lumen. Fructose is a FODMAP only when it is in excess to glucose [23,33]. When large quantities of fructose are ingested, malabsorption tends to occur in a significant proportion of individuals [34,35].

### 3.3. Fructans

Fructans are carbohydrates made up of short chains of fructose molecules, often ending with a glucose unit. When these chains contain between 2 and 9 units, they are called fructo-oligosaccharides (FOS), while chains with more than 10 units are classified as inulin [36]. Foods high in fructans are wheat including bread, pasta etc., onions, garlic, pistachio, inulin or chicory root [31]. They are poorly absorbed in the small intestine primarily due to the absence of appropriate digestive enzymes in humans [37]. As a result, they pass into the large intestine, where they are quickly fermented by gut microbiota. This fermentation process supports the growth of beneficial microbes, such as *Bifidobacteria* and *Lactobacilli* [37]. Although small amounts of fructans are typically well tolerated by individuals without gastrointestinal issues, higher intakes may lead to symptoms like acid reflux, gas, bloating, and abdominal discomfort [38]. In patients with IBS, even modest doses of fructans can provoke these symptoms to a significantly greater extent than in healthy individuals [39].

### 3.4. Galactooligosaccharides (GOS)

Galacto-oligosaccharides (GOS) are short-chain carbohydrates produced through the transgalactosylation activity of the enzyme β-galactosidase [40]. GOS are found in a variety of food products such as vegetables, pulses, nuts, dairy products, processed foods and juices [41]. They are not absorbed by humans because of the lack of enzyme α-galactosidase, which is required to break the galactosidic bonds found in compounds such us stachyose and raffinose into simpler sugars [42]. Consequently, GOS reaches the large intestine undigested, where it is quickly fermented by gut microbes [42]. In healthy individuals, this fermentation produces gas, as a normal part of digestion. However, in people with IBS, the accumulation of gas can lead to bloating, abdominal pain, and changes in bowel movement patterns. This fermentation particularly induces the growth of beneficial bacteria such as *Bifidobacteria* and *Lactobacilli* [43].

### 3.5. Polyols

Polyols are sugar alcohols that include sorbitol, mannitol, xylitol, maltitol, erythritol, isomalt, lactitol and hydrogenated starch hydrolysates according to the US FDA [44]. The most common examples are sorbitol and mannitol. Νatural food sources of polyols include certain vegetables and fruits which include sweet corn, pears, apples, blackberries, stone fruits, cauliflower, mushrooms and snow peas. Polyols are also commonly used in sugar free chewing gum, lollies and mints. They are utilized by the food industry because they have a lower calorie content per gram, do not cause dental cavities, and have minimal impact on blood sugar levels [44,45].

Excess polyol molecules, especially sorbitol and mannitol, move slowly through the small intestine, attracting water. This increases luminal water content and results in distension. Polyols that are not absorbed pass into the large intestine, where they are fermented by colonic bacteria, leading to the production of gas and causing bloating [46,47].

Table 1 presents the low-FODMAP threshold levels for each type of FODMAP sugar per serving, including oligosaccharides (a combination of fructans and galacto-oligosaccharides), polyols (such as sorbitol and mannitol), excess fructose (amounts of fructose greater than glucose), and lactose [48].

## 4. The Low-FODMAP Diet as a Therapeutic Dietary Approach

The Low-FODMAP approach is implemented in three distinct phases, ideally under the guidance of a qualified dietitian, to ensure nutritional adequacy and effective symptom management [49]. The first phase, known as the elimination phase (lasting 2–6 weeks), includes the systematic removal of all foods high in FODMAPs from diets to reduce symptom burden. This is followed by the reintroduction phase (typically lasting 6–10 weeks), in which specific high-FODMAP foods are gradually reintroduced in a controlled manner to identify specific triggers responsible for the gastrointestinal symptoms. Various reintroduction protocols for the low-FODMAP diet exist across different countries. Each food challenge can be administered over three consecutive days [50] or extended to a four-day challenge period [51]. The duration of washout periods between challenges can also vary, often depending on whether symptoms are elicited during the challenge phase [49,50,51,52]. The final maintenance/personalization phase is based on the findings from the reintroduction phase and involves the development of a tailored long-term dietary plan. This phase includes only those FODMAP-containing foods that are well tolerated by the individual, with the aim of maximizing dietary variety while minimizing symptom recurrence [49].

When patients receive education on the restriction, reintroduction, and personalization phases of the low-FODMAP diet, sustained symptom improvement has been documented in approximately 57–67% of cases [53,54]. Furthermore, up to 83% of individuals have reported a reduction of more than 50 points on the IBS Symptom Severity Scale (IBS-SSS) [51].

## 5. Effectiveness of the Low-FODMAP Diet

The effectiveness of the low-FODMAP diet in managing IBS is well-established. A meta-analysis by Peter Varjú et al. [55] evaluated the impact of the low-FODMAP diet compared to the standard IBS diet on symptom severity and quality of life in adult IBS patients, using the IBS-SSS. The analysis included data from seven randomized controlled trials (RCTs) and three prospective studies [55]. The findings confirmed that the low-FODMAP diet is more effective than the traditional IBS diet in alleviating symptoms and improving quality of life [55]. However, the study also raised concerns regarding potential alterations in gut microbiota and nutritional adequacy, particularly in the absence of guidance from a dietitian [55].

A meta-analysis by Altobelli et al. [56] confirmed that the low-FODMAP diet effectively reduces abdominal pain and bloating. Though, it has yet to be proven whether this diet offers greater long-term benefits compared to traditional IBS dietary approaches. Similar conclusions were drawn by Van Lanen et al. [57] who also emphasized concerns related to dietary adequacy and the potential impact on gut microbiota over extended periods. The superiority of the low-FODMAP diet over other dietary interventions was further supported by Ford et al. in their metanalysis which included 13 RCTs [58].

## 6. Nutritional Implications of the Low-FODMAP Diet in Patients with IBS

The reviewed body of literature provides a comprehensive overview of dietary intake changes associated with low-FODMAP diet interventions in patients diagnosed with IBS, evaluated across multiple international cohorts employing consistent diagnostic criteria (predominantly Rome III) [59,60,61,62,63]. According to the Rome III criteria (2011), IBS is characterized by recurrent abdominal pain or discomfort, occurring, more than 3 days per month in the last three months and is associated with at least two of the following features: amelioration with defecation, onset linked to a change in frequency of stool, or an onset linked to a change in stool form or appearance. Those diagnostic criteria should be met for the previous three months, with symptom onset occurring at least six months prior to diagnosis [64]. All included studies employed controlled, prospective, or cross-sectional designs, with intervention durations ranging from 3 weeks to over 2 years. Dietary intake was assessed through various methods, food diaries, Food Frequency Questionnaires (FFQ), and 24 h recalls. Such diversity provides a broad picture, although the variability in dietary assessment tools may introduce inconsistencies in nutrient intake quantification. Across studies, all major IBS subtypes (IBS-D, IBS-C, IBS-M) were represented, with a particular emphasis on IBS-D.

### 6.1. Macronutrients

Notably, the majority of interventions consistently demonstrated a reduction in total energy and carbohydrate intake in the low-FODMAP arms compared to various comparators (such as British Dietetic Association (BDA)/NICE guidelines, or habitual dietary intake). This pattern was observed in studies from Sweden [59], the USA [60], Iran [61], and China [62] suggesting a reproducible effect of the low-FODMAP diet on macronutrient consumption across diverse populations.

### 6.2. Micro-Nutrients

A critical concern with elimination diets is the potential compromise of micronutrient adequacy. Eswaran et al. [60], Ostgaard H et al. [65] and Guerreiro et al. [66] reported significant reductions in thiamin, riboflavin, calcium, iron, and sodium intake among participants following the low-FODMAP diet. Similarly, the study by Alrasheedi A. [67] documented significant reductions in micronutrients including iron, zinc, magnesium, riboflavin and vitamin C. However, some longer-term studies such as O’Keeffe et al. [53] and Ostgaard et al. [65] found higher intake of certain micronutrients (such as folate, vitamin A, B6, and β-carotene).

Overall, the evidence supports the short-term efficacy of the low-FODMAP diet in reducing caloric and carbohydrate intake, with variable effects on micronutrients. These findings reinforce the need for professional dietary guidance when implementing the low-FODMAP diet, especially beyond the initial elimination phase. Dietitian-led reintroduction phases are critical for ensuring nutritional adequacy and minimizing the risk of long-term deficiencies.

Further research is warranted to evaluate the long-term nutritional status of IBS patients on sustained or modified low-FODMAP diets, as well as to explore culturally tailored dietary strategies that maintain symptom control without compromising nutritional quality. Table 2 provides an overview of studies evaluating the nutritional implications and potential deficiencies associated with the low-FODMAP diet in IBS patients.

Evidence consistently shows that the low-FODMAP diet leads to reduced total energy and carbohydrate intake across diverse populations, as demonstrated in studies from Sweden [59], the USA [60], Iran [61], and China [62]. This reflects a reproducible trend toward lower macronutrient consumption when compared to standard dietary advice or habitual intake.

However, the restrictive nature of the diet raises concerns regarding micronutrient adequacy. Several studies have reported reductions in key nutrients such as thiamin, riboflavin, calcium, iron, zinc, magnesium, and vitamin C during the elimination phase [60,65,66,67]. On the other hand, some longer-term studies observed higher intakes of certain micronutrients—including folate, vitamin A, B6, and β-carotene [53,65].

Overall, while the low-FODMAP diet is effective for symptom control in the short term, its nutritional limitations underline the importance of professional dietary supervision. Ongoing monitoring and structured reintroduction phases are essential to ensure balanced nutrient intake and to prevent deficiencies. Further research is needed to assess the long-term nutritional status of individuals with IBS following sustained or culturally adapted low-FODMAP interventions. Table 2 provides an overview of studies evaluating the nutritional implications and potential deficiencies associated with the low-FODMAP diet in IBS patients.

## 7. Impact of the Low-FODMAP Diet on Gut Microbiota Composition in IBS Patients

The low-FODMAP diet has been widely adopted as a dietary strategy for managing IBS, particularly in individuals experiencing prominent gastrointestinal symptoms. However, beyond symptom control, growing attention has been directed toward its effects on the gut microbiome, given the central role of microbial ecology in gastrointestinal health and immune regulation [69]. The gut microbiota offers numerous advantages to the host, including strengthening the intestinal barrier by promoting tight junction integrity and preventing pathogen translocation [70] supporting the development and maintenance of the gut lining, aiding in energy extraction by fermenting dietary fibers into short-chain fatty acids (SCFAs) [71] and protecting against harmful microbes [72]. Diet plays a significant role in shaping the composition and function of the gut microbiota [73].

A variety of study designs have assessed the relationship between the low-FODMAP diet and the microbiome, including crossover [74,75] and parallel trials [63,76,77,78,79], with subject numbers ranging from 27 to 104. The duration of interventions ranged from 3 to 12 weeks, offering insights into both short- and mid-term microbiome alterations. A diversity of microbial analysis techniques was utilized, including quantitative PCR (qPCR), 16S rRNA sequencing, GA-map^TM^ Dysbiosis Test, and fluorescence in situ hybridization (FISH).

Several studies consistently reported reductions in the abundance of beneficial taxa, particularly *Bifidobacteria*. For example, Halmos et al. [74] observed a significant decrease in total bacterial abundance, with specific reductions in *Akkermansia muciniphila*, *Bifidobacteria*, and *Clostridium cluster* IV and XIVa following low-FODMAP diet intervention. Similar reductions in *Bifidobacteria* abundance were noted by other studies [75,76,77], raising concerns regarding the potential long-term consequences of such microbial shifts. *Bifidobacteria* play an important role in human health. They suppress pathogenic organisms by producing organic acids [78], antibacterial peptides [79], and quorum-sensing inhibitors [80].

Interestingly, studies utilizing 16S rRNA sequencing [63,77,81,82] generally reported no significant changes in overall α- or β-diversity. However, more granular compositional changes were noted in specific subgroups. For instance, McIntosh et al. [82] found increased Firmicutes richness in IBS-D and IBS-M patients, coupled with reduced relative abundance of *Bifidobacteria*.

While reduced *Bifidobacteria* levels are a common feature across studies, it remains uncertain whether these shifts represent clinically meaningful dysbiosis or simply a transient effect of carbohydrate restriction. Of note, Staudacher et al. [83] did not observe significant differences in the overall concentrations of key microbial groups such as Bacteroides–Prevotella and Lactobacillus–Enterococcus, suggesting a selective impact on saccharolytic bacteria.

The use of the GA-map^TM^ Dysbiosis Index in studies by Hustoft et al. [75] and Bennet et al. [76] provided a broader assessment of microbial balance. Both studies revealed that the low-FODMAP diet-treated patients experienced a worsening dysbiosis. Hustoft et al. [75] documented an increased relative abundance of Dorea, accompanied by reductions in Clostridium, Faecalibacterium prausnitzii, and other beneficial genera. These shifts raise concern for potential functional impairments in SCFA production and mucosal health, despite symptom relief.

The current evidence suggests that adherence to the low-FODMAP diet results in specific and reproducible changes in gut microbiota composition, notably a reduction in *Bifidobacteria* and other saccharolytic taxa. Although global diversity metrics may remain stable, these compositional changes warrant careful consideration, particularly due to the known health-promoting roles of these microbes.

Dietitians implementing the low-FODMAP diet should prioritize the reintroduction and personalization phases to support microbial recovery, possibly incorporating prebiotic or probiotic strategies. Further longitudinal research is needed to ascertain the long-term effects of low-FODMAP diet on microbial resilience, host metabolism, and gut barrier function.

## 8. Mediterranean Diet as a Nutritional Paradigm

The MD is not a modern dietary model, but a timeless pattern rooted in the traditional eating habits of populations in Mediterranean countries, such as Greece, Italy, and Spain. Shaped over centuries of agricultural development and cultural exchange, this diet emphasizes self-sufficiency, seasonality, and simplicity in food selection and preparation [84]. The first scientific documentation of the beneficial effects of the MD occurred in the 1960s, through the famous “Seven Countries Study,” led by Ancel Keys [85] which compared the diets and health outcomes of diverse populations. The study highlighted the low rates of cardiovascular disease in regions such as Crete and southern Italy, despite high fat consumption, primarily due to the widespread use of olive oil and low intake of animal-based saturated fats [85].

In recent years, MD has emerged as a model diet for public health due to its strong association with the prevention of cardiovascular disease [86], insulin resistance related diseases such as type 2 diabetes, obesity, metabolic dysfunction-associated steatotic liver disease (MASLD) [87], certain types of cancer [88], and neurodegenerative conditions [88,89]. Central to its health-promoting effects is the abundant consumption of plant-based foods, including fruits, vegetables, legumes, whole grains, nuts, and seeds, which provide a rich source of fiber, polyphenols, vitamins, and antioxidants with established anti-inflammatory properties [86,90,91,92,93,94,95,96].

Olive oil serves as the primary source of fat. Extra virgin olive oil is rich in monounsaturated fatty acids and phenolic compounds that protect the endothelial function and reduce oxidative stress [86,90,97]. Fish and seafood are consumed in moderate quantities, with particular emphasis on oily fish such as salmon and mackerel, which are rich in omega-3 fatty acids known for their cardioprotective and anti-inflammatory effects [86,90,91]. In contrast, red meat should be consumed sparingly, no more than twice per week, and ideally in lean cuts, while processed meats are limited to no more than one serving per week [98]. Red meat is mainly replaced by poultry, legumes, and fish, contributing to reduced intake of saturated fats and potentially harmful animal proteins [84,90]. The frequent consumption of red and processed meat has been linked to a higher risk of type 2 diabetes, cardiovascular disease, cancer and all-cause mortality [99,100].

Milk and dairy products should be included in the daily diet in moderate amounts, with no more than two servings per day [98], with a preference for traditional fermented options such as yogurt and cheese which contribute to gastrointestinal and bone health [84,90,98]. The MD also emphasizes the avoidance of industrial processed products high in trans fats, excess salt, and added sugar [84,90]. Moderate consumption of red wine, usually during meals, is a culturally embedded component of the traditional MD. While recent guidelines underscore that any amount of alcohol may pose health risks [101], the polyphenols in red wine, such as resveratrol, have been proposed to exert beneficial cardiovascular effects when consumed responsibly [84,90]. The MD also encompasses socio-cultural practices, including communal meals and mindful eating, which have been proposed to support dietary adherence and long-term sustainability [102,103]. The World Health Organization, in collaboration with Oldways and Harvard University, was among the key contributors to the first visual representation of the traditional Mediterranean Diet Pyramid (MDP), introduced in 1993 [84]. This model was later revised in 2009 and 2010 to reflect updated scientific evidence that incorporated contemporary lifestyle, dietary habits, socio-cultural factors, environmental concerns, and health challenges affecting modern Mediterranean populations [90].

The recognition of the MD further in 2010, when UNESCO included it in the Representative List of the Intangible Cultural Heritage of Humanity, acknowledging its role as both a health-promoting and socio-cultural phenomenon [104]. In 2011, the FAO and CIHEAM evaluated the MD model across four dimensions, health and nutrition, environmental impact, economic resilience and socio-cultural values leading to the development of practical sustainability indicators, including thirteen nutritional metrics [105]. Subsequent efforts, such as the establishment of the International Mediterranean Diet Foundation (IFMeD) in 2014 [106] and the presentation of the Med Diet 4.0 framework at EXPO Milan in 2015, reinforced the integrated benefits across public health, environment, economic stability, and culture. Continued scientific consultation led to the revisions of the MD pyramid, incorporating updated environmental and health data with the 2020 update emphasizing environmental sustainability—promoting reduced red meat and bovine dairy consumption and encouraging legumes and locally grown plant-based foods [98,106]. This evolution reflects a growing consensus that dietary patterns must address not only individual well-being but also planetary health.

## 9. Interventional Studies Investigating the Mediterranean Diet in IBS

Paduano et al. [107] conducted a 12-week clinical trial comparing the effects of a balanced MD, a low-FODMAP diet and a gluten-free diet in patients diagnosed with IBS according to Rome IV criteria. All three dietary interventions significantly reduced IBS symptom severity. Notably, the MD demonstrated comparable efficacy to the low-FODMAP diet LFD (symptom score LFD: 16 ± 8 vs. MD: 17 ± 7, *p* = 0.44). The MD also showed the highest patient acceptance, potentially due to its structured meal pattern and the inclusion of both FODMAP containing foods and gluten, highlighting the importance of meal distribution in symptom relief [96].

In a six-week randomized Australian study by Staudacher et al. [108], 83% of participants following the MD demonstrated a significantly higher response rate in symptom improvement, compared to 37% in the control group (*p* < 0.001). Despite similar FODMAP intake and macronutrient distribution between groups, the MD group consumed significantly more monounsaturated fats (*p* = 0.041) and less sorbitol (*p* < 0.05). This suggests that qualitative dietary factors may play a role in symptom outcomes independently of FODMAP restriction [108].

Kasti et al. [109] further supported the idea of an integrated approach as they found that the combination of the MD with a low-FODMAP strategy (MED–LFD) resulted in significantly greater improvements in symptom severity (*p* ≤ 0.001), higher responder rates (*p* = 0.007), and better dietary adherence (*p* = 0.007) compared to the group following NICE dietary guidelines. These findings suggest that the MED–LFD diet is a more effective and sustainable dietary intervention for managing IBS symptoms [109].

## 10. Observational and Cross-Sectional Evidence

Several observational and cross-sectional studies have examined the association between adherence to the MD and symptom patterns in individuals with IBS. Zito et al. and Altomare et al. reported significantly lower MD adherence among IBS patients compared to controls [9,110]. Zito et al. conducted a large-scale observational study involving 1193 participants, revealing markedly reduced MD adherence score in the IBS group (*p* < 0.001) [110]. Similarly, Altomare et al. identified dietary imbalances in IBS patients, notably lower intakes of vegetables, nuts, dairy, and seafood among IBS patients (*p* < 0.05 for each group) suggesting potential nutritional inadequacies related to symptom burden [9].

Further evidence from a recent case–control study by Baghdadi et al. in Iran demonstrated that higher adherence to the MD was associated with a 51% reduction in IBS odds (OR: 0.49; 95% CI: 0.30–0.73, *p* < 0.001) [111]. Participants in the high-adherence group reported increased consumption of fiber, legumes, nuts, whole grains, and seafood, and lower intakes of processed foods and saturated fats, an overall dietary profile consistent with reduced pro-inflammatory potential and improved gut function.

On the other hand, in a U.S.-based cohort study, Chen et al. explored MD adherence using both the alternate Mediterranean Diet Score (aMED) and the Mediterranean Diet Adherence Screener (MEDAS) [112]. While no significant differences in overall MD adherence were reported between IBS and controls, higher consumption of some anti-MD-aligned food groups (soda, processed meat, baked goods, and beer) was correlated with lower symptom severity [112]. In contrast, pro-MD food consumption (cantaloupe, carrot juice, grapefruit, sweet potato, and oranges/tangerines/clementines) was associated with increased symptom scores [112]. Interestingly, a higher adherence to a symptom-modified MD pattern was also correlated with favorable changes in gut microbiota, including reductions in potentially harmful genera such as *Faecalitalea* and *Streptococcus*. Chen et al. concluded that an IBS-adapted MD, rather than the standard approach, may be more effective in reducing symptom severity and should be considered in both research and clinical practice [112].

Collectively, current observational evidence suggests that the MD may represent a promising dietary strategy for managing IBS symptoms. It appears to exert beneficial effects not only through its nutrient composition, high in fiber, antioxidants, and unsaturated fats, but also from enhanced meal regularity and positive effects on gut microbial population. Importantly, its inclusive and culturally adaptable nature may promote long-term adherence, addressing a key limitation of more restrictive diets such as the low-FODMAP diet. Table 3 summarizes key interventional and observational studies examining the impact of the Mediterranean Diet on IBS symptomatology and adherence.

Nonetheless, limitations in the existing literature must be acknowledged. While findings from controlled trials are encouraging, heterogeneity in study design, population characteristics and adherence measurement tools limit direct comparisons across studies. Moreover, observational data reveal inconsistent associations between overall MD adherence and IBS symptomatology, possibly due to individual variability in food tolerance and gut sensitivity, underscoring the need for personalized dietary approaches in IBS care. Considering individual FODMAP intolerances, a modified Mediterranean Diet tailored to the patient’s specific sensitivities may offer improved symptom management, further supporting the need for personalized nutritional strategies in IBS care.

## 11. Addressing Nutritional Inadequacies Through the Mediterranean Diet

One of the main concerns associated with restrictive diets like the low-FODMAP diet is the risk of nutritional deficiencies, particularly in fiber, iron, calcium, zinc, magnesium, B vitamins (e.g., riboflavin and thiamin), and vitamin C [60,65,66,67,68]. These deficiencies often result from the elimination of nutrient-dense foods such as legumes, dairy, whole grains, and certain fruits and vegetables [113]. In contrast, the MD is naturally rich in micronutrients, and its inclusion of diverse plant-based foods, healthy fats, lean proteins, and moderate dairy can help counteract these potential deficits [12].

Fiber intake, in particular, has consistently been shown to decrease in individuals adhering to a low-FODMAP diet. Most studies reported a reduced or insufficient intake of carbohydrates and especially dietary fiber among low-FODMAP groups [59,60,61,62,63,66,67]. This is concerning given the role of non-digestible carbohydrates, especially soluble fibers such as fructooligosaccharides (FOS), galactooligosaccharides (GOS), inulin, and fructans, as prebiotics with recognized anti-inflammatory properties [114]. Moreover, these substrates promote the production of SCFAs, such as butyrate, which are instrumental in preserving intestinal epithelial barrier integrity and modulating both mucosal and systemic immune responses [115]. The MD is rich in prebiotic-containing foods, particularly dietary fibers and complex carbohydrates (legumes, whole grains, fruits, vegetables), that serve as substrates for beneficial gut microbes [12]. These foods contain FOS, inulin, GOS, and resistant starches, which selectively promote the growth of *Bifidobacteria* and *Lactobacilli*, improve SCFA production (notably butyrate), and support intestinal barrier function [116]. In addition to FODMAP-associated prebiotics, the MD includes a wide range of foods rich in non-FODMAP dietary fibers. These fibers are characterized by low fermentability, low osmotic activity, viscosity and gel-forming properties, and insolubility, qualities that contribute to better gut tolerance, especially in individuals with IBS [117]. Notable examples include β-glucans found in oats and barley [118], cellulose and certain hemicelluloses present in vegetables like spinach, kale, carrots, zucchini, green beans, oranges, almonds, walnuts and whole grains such as brown rice [119]. The diet also includes resistant starch, found in foods like green bananas, legumes, peas and cooked and cooled starch foods [120]. These fibers support gut health by promoting healthy transit, enhancing microbial diversity, and improving stool form, all while minimizing symptoms typically associated with high-FODMAP fermentable fibers [118]. Emerging evidence also suggests that adherence to the MD is linked to lower dysbiosis scores and favorable microbial shifts even in IBS populations [112]. Compared to the LFD, which focuses on symptom suppression through carbohydrate restriction, the MD offers a microbiome-supportive dietary pattern that may address underlying dysbiosis rather than only its symptomatic manifestations [121,122].

In terms of calcium and riboflavin, IBS patients usually avoid milk and dairy products [9,65] due to their lactose intolerance [123]. In Western populations, milk and dairy products represent the primary source of dietary calcium, accounting for approximately 50–75% of total daily calcium intake [124]. Adequate intake of calcium is supported in MD through moderate consumption of dairy products like yogurt and cheese [106]. Fermented dairy products serve as sources of probiotics that help maintain a healthy balance of gut microbiota, supporting immune function and helping reduce inflammation [125,126]. A prospective study on 189 patients with IBS reported complete symptom remission in individuals consuming homemade yogurt, suggesting potential therapeutic effects [127]. Moreover, GOS present in many MD foods, have been shown to enhance calcium absorption [128,129], while dairy products are a key source of riboflavin, accounting for approximately 25–30% of the total intake in typical Western diets [124,130].

Iron and vitamin C intake may also be compromised under low-FODMAP diet, particularly due to the restriction of various fruits and vegetables, rich in flavonoids, carotenoids, vitamin C, phenolic acids, and anthocyanins [53,131]. In particular, the decreased consumption of vitamin C-rich foods, alongside the elimination of carbohydrate sources high in fructans and GOS, can further impair iron absorption and intake [132,133]. Conversely, the MD, which includes iron-rich staples such as legumes, nuts, seeds, and seafood, contributes to improved iron status [84,90]. Vitamin C, which enhances nonheme iron absorption, is abundantly found in fresh fruits and vegetables like citrus fruits, tomatoes, and peers [84,90,134].

Beyond its high nutrient density, the MD promotes structured eating patterns and regular meal timing. Several studies have reported that individuals adhering to a low-FODMAP diet often reduce their meal frequency, which may compromise both nutrient intake and gastrointestinal stability [59,60]. In contrast, Paduano et al. [107] observed that patients following a balanced Mediterranean-style regimen, characterized by five evenly distributed meals (breakfast, mid-morning snack, lunch, mid-afternoon snack, and dinner), experienced significant symptom improvement. The authors attributed this benefit to the redistribution of FODMAP-containing foods across multiple meals, thereby minimizing luminal osmotic load and fermentative stress in the gut, resulting in better digestive tolerance. This structured meal pattern may also facilitate more efficient nutrient absorption and attenuate postprandial gastrointestinal distress, underscoring the importance of meal patterning in dietary management of IBS.

Taken together, these observations invite a broader rethinking of IBS not as an invariably chronic condition, but as a potentially reversible disorder in certain individuals. Through sustained dietary modulation of the gut microbiota, particularly via Mediterranean-style low-FODMAP approaches, there is increasing potential to not only alleviate symptoms, but to restore gut microbial diversity, reduce mucosal inflammation, and reestablish barrier integrity [109,135]. If such microbiota-focused dietary interventions are maintained over time, they may contribute to long-term remission or even full symptom resolution in a subset of patients. Although additional long-term studies are necessary to fully establish this therapeutic potential, these findings highlight a critical shift in IBS care: from short-term symptom control to the prospect of modifying disease trajectory through microbiota-supportive nutrition.

## 12. Conclusions

In light of the current evidence, integrating principles of the MD into the framework of the low-FODMAP diet emerges as a promising strategy for the comprehensive management of IBS. While the low-FODMAP diet remains effective for symptom control, especially during its elimination phase, concerns regarding its nutritional adequacy, long-term sustainability, and potential microbiome disruption cannot be overlooked. The MD, with its anti-inflammatory profile, high nutrient density, and gut microbiota-supportive properties, offers a complementary dietary pattern that can mitigate these limitations.

By introducing MD components during the low-FODMAP diet, patients can achieve a more balanced and adaptable dietary model. This integrative approach supports dietary diversity, enhances intake of key micronutrients such as calcium, iron, and B vitamins, and fosters the restoration of a healthy gut microbial ecosystem through the inclusion of prebiotic fibers, polyphenols, and fermented foods. Clinical and observational studies increasingly suggest that the MD is not only well-tolerated by IBS patients but may also enhance adherence and long-term outcomes through its structured meal pattern and sensory appeal. However, further randomized trials with standardized adherence metrics and long-term follow-up are warranted to validate its efficacy and refine personalized implementation strategies. Ultimately, a Mediterranean-style low-FODMAP diet may represent a paradigm shift in IBS dietary management, addressing not only symptom suppression but also nutritional adequacy, microbiota resilience, and sustainable health promotion.

Future studies should focus on refining implementation protocols, assessing long-term effects, and identifying patient-specific predictors of success in integrating the Mediterranean Diet with the low-FODMAP approach.

## Figures and Tables

**Table 1 microorganisms-13-02085-t001:** Threshold Levels of FODMAPs per Serving.

Individual FODMAPs	Grams Per Serve ^α^ (Individual Food)
Oligosaccharides ^β^ (core grain products, legumes, nuts, and seeds)	<0.30
Oligosaccharides ^β^ (vegetables, fruits, and all other products)	<0.20
Polyols—sorbitol or mannitol	<0.20
Total polyols	<0.40
Excess fructose ^γ^	<0.15
Excess fructose (for fresh fruit and vegetables when “fructose in excess of glucose” is the only FODMAP present)	<0.40
Lactose	<1.00

Notes: ^α^: Standard serve size; ^β^: Oligosaccharides = total fructans plus galacto-oligosaccharides (stachyose and raffinose); ^γ^: Excess fructose = fructose − glucose.

**Table 2 microorganisms-13-02085-t002:** Low-FODMAP Diet and Nutritional Deficiencies.

Study	Duration	Study Design	Criteria Used for IBS Diagnosis and IBS Subtype	Methodology	Results
Bohn 2015 [59], Sweden	4 weeks	38 patients on LFD 37 patients on BDA/NICE diet	Rome III IBS-C IBS-D IBS-M	4-day food diary (at screening and during last week of intervention period)	LFD group had greater reductions in energy (*p* = 0.03), carbohydrates (*p* = 0.007), dietary fiber (*p* = 0.003) and meals/day (*p* = 0.05) vs. BDA/NICE diet
Eswaran 2019 [60], USA	4 weeks	41 patients on LFD 39 patients on mNICE diet	Rome III IBS-D	3-day food diary (at baseline and last week of intervention period).	Reduction in energy, carbs and number of meals in both groups (*p* < 0.01). Reduction in thiamin (*p* < 0.01), rivoflabin (*p* < 0.05), calcium (*p* < 0.01) and sodium (*p* < 0.001) in the LFD vs. mNICe group. After calorie-adjustment: reduction in total sugars (*p* < 0.001), carbs and sodium (*p* < 0.01), rivoflabin (*p* < 0.05) in the LFD group vs. mNICE. Fewer patients met the DRIs for thiamin and iron in the LFD group, vs. fewer patients meeting the DRIs for calcium and copper in the control group.
Zahedi 2018 [61], Iran	6 weeks	55 patients on LFD 55 patients on BDA diet	Rome III IBS-D	3-day food diary (at baseline and last week of intervention period)	Reduced energy (*p* < 0.05) and carbs for both groups with greater carb reduction in LFD (*p* < 0.001) vs. BDA (*p* < 0.05). Fat reduction in BDA group (*p* < 0.001).
Zhang 2021 [62], China	3 weeks	54 patients on LFD 54 patients on BDA/NICE diet	ROME III IBS-D	3-week food diary	Reduced total intake of energy and carbs in LFD vs. BDA/NICE group (*p* < 0.05)
Harvie 2017 [63], New Zealand	3 months	23 patients on LFD 27 patients on habitual diet	Rome III IBS-C IBS-D IBS-M	FFQ	Energy reduction in both groups (LFD, *p* < 0.01, Hab diet, *p* < 0.05) and fiber reduction from baseline to 3 months in LFD (*p* < 0.01)
Guerreiro 2020 [66], England	4 weeks	47 patients on LFD 23 patients on BDA	ROME IV IBS-C IBS-D IBS-M	24 h dietary recall method (baseline, 4th week follow-up, 10th week follow-up	Both groups had significant reductions from baseline to 4 weeks in energy (LFD, *p* = 0.001) carb intake (*p* = 0.006) and LFD had significant reduction in total fiber (*p* = 0.048) and iron (*p* = 0.003) compared to BDA group
O’Keeffe 2018 [53], UK	Prospective 6–18-month follow-up after initial 6-week LFD	84 patients on LFD, 19 patients on habitual diet	NICE	Semi-quantitative FFQ (at follow-up)	Higher folate (*p* = 0.02) and vitamin A (*p* = 0.045) intakes compared to habitual diet
Ostgaard 2012 [65], Norway	Prospective follow-up study 2 years after LFD advice	35 controls, 36 unguided IBS patients (no advice), 43 guided IBS patients (LFD advice)	ROME III	MoBa FFQ	No statistically significant difference for energy and macronutrient intake between groups. Lower intakes of riboflavin and calcium (*p* < 0.05). Higher intake of β-carotene and vitamin B6 (*p* < 0.05) for LFD guided patients vs. healthy controls.
Pourmand 2018 [68], Iran	Cross-sectional	3362 IBS patients Quintiles of FODMAP adherence	A modified Persian version of ROME III questionnaire IBS-C IBS-D IBS-M	106-item semi-quantitative food frequency questionnaire	Individuals with the highest adherence to the LFD had lower dietary intakes of all measured foods groups and (micro) nutrients (*p* < 0.001)
Alrasheedi 2025 [67], Saudi Arabia	10-week intervention program on LFD	45 IBS patients (baseline on LFD)	ROME III IBSD IBSM	FFQ	Energy reduction (*p* < 0.01). Reductions in carbs, fiber, starch, sugar, fat, iron, zinc, magnesium, sodium, potassium, riboflavin, and vitamin C.

Abbreviations: IBS, Irritable Bowel Syndrome; IBS-C, IBS with predominant Constipation; IBS-D, IBS with predominant Diarrhea; IBS-M, IBS with Mixed bowel habits; IBS-U, IBS Unclassified; LFD, Low-FODMAP Diet; MD, Mediterranean Diet; DRIs, Dietary Reference Intakes; FFQ, Food Frequency Questionnaire; MoBa FFQ, Norwegian Mother and Child Cohort Food Frequency Questionnaire; NICE, National Institute for Health and Care Excellence; BDA, British Dietetic Association; mNICE, modified NICE diet; ROME III/IV, Rome III/Rome IV Diagnostic Criteria; SCFAs, Short-Chain Fatty Acids; MUFA, Monounsaturated Fatty Acids; GOS, Galactooligosaccharides; FOS, Fructooligosaccharides.

**Table 3 microorganisms-13-02085-t003:** Mediterranean Diet and Irritable Bowel Syndrome.

Study	Duration	Study Design	Criteria Used for IBS Diagnosis	Methodology	Result
Paduano 2019 [107], Italy	12 weeks	Clinical trial 28 patients on balanced Med diet 30 patients on gluten free diet 34 patients on low-FODMAP diet	ROME IV IBS-C IBS-D IBS-M IBS-U	24 h food recall	All three diets reduced the severity of symptoms (*p* < 0.01). The MD was as effective as the LFD with no difference between them (*p* = 0.44). The MD had a higher acceptance rate.
Staudacher 2024 [108], Australia	6 weeks	Clinical trial 29 patients on MD 30 patients on habitual diet	ROME IV IBS-C IBS-D IBS-M IBS-U	MEDAS DSS 3-Day Food Diary	Energy, macronutrient and fiber intake were not different. Contribution of MUFAs to total energy was higher in MD compared with controls (*p* = 0.041). There was a greater proportion of responders to gastrointestinal symptoms in the MD group than the control group (*p* < 0.001).
Kasti A 2025 [109], Greece	6 months	Clinical trial 54 patients on MD-LFD 54 patients on NICE dietary guidelines	ROME IV IBS-D IBS-M IBS-U	5-point Likert scale on the adherence to the diet through weekly telephone	The MD–LFD group showed a significantly greater improvement in symptom severity (*p* < 0.001) and (*p* = 0.001). Responder rates (84.6% vs. 60.8%, *p* = 0.007) and (79.1% vs. 52.3%, *p* = 0.006). Adherence (75% vs. 41%, *p* = 0.007) and 45% vs. 7%, *p* < 0.001).
Zito 2016 [110], Italy	May 2011–April 2012	Observational study 1193 patients 172 IBS 719 controls 243 FD	ROME ΙΙΙ	FFQ KIDMED for 17–24 aged patients SMDQ to those older than 24 years	A significantly lower MD adherence score was found in individuals with IBS compared to controls (*p* < 0.001)
Chen 2024 [112], USA	July 2013–November 2021	Cross-Sectional Study 106 IBS 108 controls	Rome III and Rome IV IBS-C IBS-D IBS-M IBS-U	Diet history Questionnaire II aMED MEDAS Healthy Eating Index-2010	Adherence to the MD was similar between IBS and control subjects and was not associated with IBS-SSS. There were no differences in aMED and MEDAS scores between IBS and control (*p* = 0.83) and (*p* = 0.46).
Altomare 2021 [9], Italy	2015–2017	Cross-Sectional Study 28 IBS 21 controls	Rome IV IBS-C IBS-D IBS-M	3-Day Food record FFQ	Adherence to the MD Med score was lower in the IBS group compared to the control group *p* < 0.01
Baghdadi 2025 [111], Iran	December 2023–June 2024	Case–control Study 170 IBS 340 controls	Rome IV IBS-C IBS-D IBS-M IBS-U	Semiquantitative 168-item FFQ MD PDQS-dietary quality index	Greater adherence to the MD was associated with a 51% lower likelihood of IBS (*p* < 0.001). Participants in the higher quartile of MD and PDQS significantly consumed lower amounts of energy and macronutrients compared to subjects in the first quartile (*p* < 0.001).

Abbreviations: IBS, Irritable Bowel Syndrome; IBS-C, IBS with predominant Constipation; IBS-D, IBS with predominant Diarrhea; IBS-M, IBS with Mixed bowel habits; IBS-U, IBS Unclassified; MD, Mediterranean Diet; LFD, Low-FODMAP Diet; MED-LFD, Mediterranean-style Low-FODMAP Diet; NICE, National Institute for Health and Care Excellence; FD, Functional Dyspepsia; ROME III/IV, Rome III/Rome IV Diagnostic Criteria for functional gastrointestinal disorders; FFQ, Food Frequency Questionnaire; KIDMED, Mediterranean Diet Quality Index for Children and Adolescents; SMDQ, Short Mediterranean Diet Questionnaire; MEDAS, Mediterranean Diet Adherence Screener; DSS, Diet Satisfaction Score; aMED, Alternate Mediterranean Diet Score; PDQS, Prime Diet Quality Score.

## Data Availability

No new data were created or analyzed in this study. Data sharing is not applicable to this article.

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
