# Peer review of "Optimizing Nutritional Balance: Integrating the Mediterranean Diet into Low-FODMAP Nutrition"

_microorganisms, 2025, doi:10.3390/microorganisms13092085_

Round 1

Reviewer 1 Report

Comments and Suggestions for Authors

The topic of the article fits within the scope of the journal in terms of food microbiology, but it would fit also in Journal Foods. The article is well written and covers the specific aspect of Mediterranean Diet and Low FODMAP Nutrition. There are some minor comments that need to be addressed.

Please rewrite the first three sentences of the abstract, due to high similarity with the parts in the Introduction.

Page 2 Line 79. acronym FODMAP is already defined earlier, please correct.

Page 2 Line 80. Please add which specific polyols are classified as FODMAPs, also in what circumstances the fructose is classified as FODMAPs. Not all Fructans are FODMAPs, more precisely the Fructooligosaccharides are classified as FODMAPs, higher fructans are not FODMAPs.

Page2 2-3 Lines 91-100. Sentences are from the same reference please add another one or just write (14) in line 100.

Page 3 lines 127-133. Please add the explanation that the Fructose is the FODMAP only when it is in the excess to glucose.

At the end of section 3 there should be small table with benchmark or cut-off values for each FODMAP compound category.

Page 5 Line 217. If Rome IV criteria was explained the Rome III criteria must also.

Author Response

The topic of the article fits within the scope of the journal in terms of food microbiology, but it would fit also in Journal Foods. The article is well written and covers the specific aspect of Mediterranean Diet and Low FODMAP Nutrition. There are some minor comments that need to be addressed

Response: Thank you for your positive evaluation and for acknowledging the relevance and quality of the manuscript. We appreciate your suggestion regarding the journal fit and are pleased to hear that the topic aligns well with the scope. All minor comments have been carefully addressed, and corresponding revisions have been made to strengthen the manuscript further.

Comment 1: Please rewrite the first three sentences of the abstract, due to high similarity with the parts in the Introduction

Response to comment 1: Thank you for your observation. The first three sentences of the abstract have been rewritten to ensure originality and reduce overlap with the Introduction, while maintaining clarity and relevance to the study's objectives (page 1, lines 10-13).

Comment 2: Page 2 Line 79. acronym FODMAP is already defined earlier, please correct.

Response to comment 2: We have revised as suggested (page 3, lines 114).  

Comment 3: Page 2 Line 80. Please add which specific polyols are classified as FODMAPs, also in what circumstances the fructose is classified as FODMAPs. Not all Fructans are FODMAPs, more precisely the Fructooligosaccharides are classified as FODMAPs, higher fructans are not FODMAPs.

Response to comment 3: Thank you for your valuable input. We have made changes and additions to the relevant section to improve clarity and enhance the reader’s understanding of the classification of FODMAP components (page 3, lines 114-117).

Comment 4: Page2 2-3 Lines 91-100. Sentences are from the same reference please add another one or just write (14) in line 100.

Response to comment 4: We revised accordingly, as suggested.

Comment 5: Page 3 lines 127-133. Please add the explanation that the Fructose is the FODMAP only when it is in the excess to glucose.

Response to comment 5: We have revised as suggested (page 4, line 178).

Comment 6: At the end of section 3 there should be small table with benchmark or cut-off values for each FODMAP compound category.

Response to comment 6: We have revised as suggested (page 5, table 1).

Comment 7: Page 5 Line 217. If Rome IV criteria was explained the Rome III criteria must also.

Response to comment 7: We have revised as suggested (page 6, lines 280-286).

Reviewer 2 Report

Comments and Suggestions for Authors

Your idea is good enough, although the originality is reduced by an original work by Kasti et al. (2025), already cited by you. The cited article is not a proposal but a proved fact.

Lines 24-27. “The aim is to propose a nutritionally adequate, sustainable, and symptom-effective model that preserves the anti-inflammatory and gut-supportive benefits of the MD. Practical, evidence-based recommendations for a modified Mediterranean-style low-FODMAP diet tailored to individuals with IBS are also presented”. The aim by Kasti et al. (2025): “To evaluate the effectiveness of the Mediterranean version of the low-FODMAP Diet (MED–LFD) compared to NICE guidelines for IBS…”. And they conclude that: “The MED–LFD is superior to NICE recommendations in managing non-constipated IBS symptoms and quality of life”.

There are several points to be reviewed.

Do not repeat the FODMAP acronym along the manuscript, it was written in lines 13, 44-45, and 79. Just one is enough.

In line 11, you have that up to 10-15% of the global population suffer IBS; however, in lines 38-39 the information about prevalence is less than 6% in different populations. What is the right information?.

Lines 94-97 about the mechanisms of IBS damage. Your manuscript affirm: “This osmotic activity draws water into the lumen, increasing stool volume and accelerating intestinal transit, which can lead to diarrhea in susceptible individuals. Furthermore, once FODMAPs reach the colon, they undergo rapid fermentation by resident colonic bacteria, producing gases such as hydrogen and methane.” Nobody is able to absorb an excess of fructose or to digest several FODMAP as fructans for instance, and they do not suffer diarrhea. Additionally, what about the constipation IBS (in American populations it is majority)?. Please look for an explanation based in dysbiosis of microbiota or so.

In subtitle 3.1 for lactose you followed a right order: described the chemical composition (a disaccharide) and supplied examples of foods rich in lactose. However, in subtitles 3.2, 3.3, and 3.4 you forgot the order without chemical structure or foods examples. It could be good to follow the same order than for lactose.

You are missing information about the breath test looking for FODMAP intolerance in IBS patients. Breath testing for carbohydrate intolerance is already standardized and essential for the diagnosis and management of IBS. Instead to try for several months, the right low-FODMAP diet, it personalized low-FODMAP diet can be introduced just after tests. Perhaps you do not agree about it, but is part of the information published the last 10 years or more. According, the low-FODMAP diet can be well designed covering macro- and micronutrients if the particular intolerances are known after breath tests. Few IBS patients are intolerant to each FODMAP type but 2 or 3 of them.

Try to re-design Tables 1 and 2. In the current presentation is not easy to follow because the interlines, fond size and type. Try to include information of columns second and third in Table 1 in other columns; for instance, cite and country in the first column, and duration in the Methods column.  

Lines 279-281. Your posterior discussion is good, you do not need to have the same methods to talk about the effect of diet in microbiota.

Subtitle 9. After considering the particular FODMAP intolerances, the MD could be even better designed for IBS patients, in agreement with your affirmation in line 451.

Finally, there are two concepts to look for: What about the inclusion of non-FODMAP dietary fiber? there are several publications about. and what about the total recovering after to modify the microbiota by a right diet?. IBS is not forever.

Author Response

Reviewer 2

Comment 1: Your idea is good enough, although the originality is reduced by an original work by Kasti et al. (2025), already cited by you. The cited article is not a proposal but a proved fact.

Lines 24-27. “The aim is to propose a nutritionally adequate, sustainable, and symptom-effective model that preserves the anti-inflammatory and gut-supportive benefits of the MD. Practical, evidence-based recommendations for a modified Mediterranean-style low-FODMAP diet tailored to individuals with IBS are also presented”. The aim by Kasti et al. (2025): “To evaluate the effectiveness of the Mediterranean version of the low-FODMAP Diet (MED–LFD) compared to NICE guidelines for IBS…”. And they conclude that: “The MED–LFD is superior to NICE recommendations in managing non-constipated IBS symptoms and quality of life”.

Response to comment 1: Thank you for your valuable feedback. We acknowledge that the study by Kasti et al. (2025) represents a significant advancement in the integration of the Mediterranean diet with the low-FODMAP protocol and that it provides evidence supporting the clinical efficacy of this combined approach. While our manuscript cites and builds upon their findings, our work differs in scope and objective. Specifically, we aim to provide a narrative review that contextualizes existing evidence and develops practical, evidence-informed dietary guidance rather than presenting new clinical outcomes. In response, we have clarified the aim and scope of our manuscript in the abstract to emphasize that this work is a review and synthesis of the literature, with a focus on nutritional adequacy, sustainability, and practical application, rather than proposing a new model or evaluating its efficacy (page 1, lines 26-30).

Comment 2: Do not repeat the FODMAP acronym along the manuscript, it was written in lines 13, 44-45, and 79. Just one is enough.

Response to comment 2: We have revised as suggested.

Comment 3: In line 11, you have that up to 10-15% of the global population suffer IBS; however, in lines 38-39 the information about prevalence is less than 6% in different populations. What is the right information?.

Response to comment 3: Thank you for highlighting this inconsistency. The discrepancy in prevalence rates has been clarified by specifying that variations are due to differences in diagnostic criteria (e.g., Rome III vs. Rome IV). The text has been revised to reflect this distinction and ensure consistency throughout the manuscript (page 1, lines 12-13).

Comment 4: Lines 94-97 about the mechanisms of IBS damage. Your manuscript affirm: “This osmotic activity draws water into the lumen, increasing stool volume and accelerating intestinal transit, which can lead to diarrhea in susceptible individuals. Furthermore, once FODMAPs reach the colon, they undergo rapid fermentation by resident colonic bacteria, producing gases such as hydrogen and methane.” Nobody is able to absorb an excess of fructose or to digest several FODMAP as fructans for instance, and they do not suffer diarrhea. Additionally, what about the constipation IBS (in American populations it is majority)?. Please look for an explanation based in dysbiosis of microbiota or so.

Response to comment 4: Thank you for your insightful comment. We agree that the pathophysiology of IBS is complex and cannot be solely attributed to osmotic or fermentative effects of FODMAPs. To address this, we have added a dedicated paragraph earlier in the manuscript discussing the multifactorial etiology of IBS (page 2, lines 49-57).  This broader context allows for a more comprehensive understanding of symptom variability across IBS subtypes, including both diarrhea-predominant (IBS-D) and constipation-predominant (IBS-C) forms. Additionally, there is an expanded explanation about FODMAPs and IBS-C (page 3, lines 136-140). 

Comment 5: In subtitle 3.1 for lactose you followed a right order: described the chemical composition (a disaccharide) and supplied examples of foods rich in lactose. However, in subtitles 3.2, 3.3, and 3.4 you forgot the order without chemical structure or foods examples. It could be good to follow the same order than for lactose.

Response to comment 5: We have revised as suggested.

Comment 6: You are missing information about the breath test looking for FODMAP intolerance in IBS patients. Breath testing for carbohydrate intolerance is already standardized and essential for the diagnosis and management of IBS. Instead to try for several months, the right low-FODMAP diet, it personalized low-FODMAP diet can be introduced just after tests. Perhaps you do not agree about it, but is part of the information published the last 10 years or more. According, the low-FODMAP diet can be well designed covering macro- and micronutrients if the particular intolerances are known after breath tests. Few IBS patients are intolerant to each FODMAP type but 2 or 3 of them

Response to comment 6: Thank you for raising this important point. We acknowledge the role that hydrogen and methane breath tests (BTs) can play in identifying specific carbohydrate intolerances. While these tests are standardized and used in clinical practice, especially for diagnosing monosaccharide and disaccharide malabsorption, their utility in guiding the full spectrum of low-FODMAP dietary personalization remains an area of ongoing discussion.

Current evidence suggests that breath testing can help identify individual FODMAP intolerances and, when used appropriately, may support a more targeted and efficient reintroduction phase. However, recent international guidelines (e.g., from the British Society of Gastroenterology, American Gastroenterological Association, NICE and Monash University) do not recommend breath testing as a routine prerequisite for initiating the low-FODMAP diet due to variability in test sensitivity, specificity, and standardization across FODMAP subtypes beyond lactose and fructose.

We chose not to include it in this review, as it falls outside the specific scope and objectives of the manuscript. Our focus is on dietary strategies, particularly the integration of the Mediterranean Diet with the low-FODMAP approach, rather than diagnostic tools. Moreover, given the ongoing debate regarding the clinical utility of breath testing for broader FODMAP subtypes, and the current lack of consensus in international guidelines, we consider its inclusion beyond the intended framework of our article.

Comment 7: Try to re-design Tables 1 and 2. In the current presentation is not easy to follow because the interlines, fond size and type. Try to include information of columns second and third in Table 1 in other columns; for instance, cite and country in the first column, and duration in the Methods column.  

Response to comment 7: Thank you for your helpful suggestion. Tables 1 and 2 have been redesigned to improve readability. Additionally, as advised, we integrated the “Author” and “Country” information into the first column for clarity. While we did not merge “Duration” with “Methodology,” we instead included “Diagnostic Criteria” and “IBS Subtypes” together to better reflect the clinical context and support comparative interpretation across studies. We believe this format enhances both structure and accessibility for the reader.

Comment 8: Lines 279-281. Your posterior discussion is good, you do not need to have the same methods to talk about the effect of diet in microbiota

Response to comment 8: Thank you for your thoughtful observation. We have revised the text accordingly (page 12, lines 365-367).

Comment 9: Subtitle 9. After considering the particular FODMAP intolerances, the MD could be even better designed for IBS patients, in agreement with your affirmation in line 451.

Response to comment 9: Thank you for your comment. As suggested, we have revised Subtitle 9 and aligned the content more closely with the statement in line 451 to emphasize that, after considering individual FODMAP intolerances, the Mediterranean Diet can be further optimized for IBS patients (page 15, lines 542-544).

Comment 10: Finally, there are two concepts to look for: What about the inclusion of non-FODMAP dietary fiber? there are several publications about. and what about the total recovering after to modify the microbiota by a right diet?. IBS is not forever.

Response to comment 10: Thank you for your observation. As far as, non-FODMAP dietary fibers in the MD is concerned, we have expanded the discussion to include their relevance for IBS management (page 20, lines 573-583).

As for the total recovering, we fully agree that IBS, particularly in its functional forms, may not always be a lifelong condition, and that remission is indeed possible. In response to your suggestion, we have added a brief but important clarification in Section 11. Specifically, we now highlight that long-term adherence to a personalized Mediterranean-style diet, which supports microbial diversity and intestinal barrier function, may promote not only sustained symptom improvement but also potential recovery in selected individuals. While more longitudinal research is needed, these insights align with the evolving view that IBS is a dynamic condition influenced by modifiable factors such as diet and microbiota status (page 21, lines 622-632).

Reviewer 3 Report

Comments and Suggestions for Authors

The present paper by Dimitriou et al. addresses a very important problem about the integration of the mediterranean diet into the low FODMAP dietary framework for managing Irritable Bowel Syndrome. The work is comprehensive, well-structured, and is a robust review of the existing literature.

In general, the manuscript is well-written and logically organized, and with strong sectioning.  The literature review is up-to-date. The described low FODMAP model is innovative and supported by preliminary clinical evidence. The paper highlights clinical, nutritional, and microbiological considerations, offering a truly multidisciplinary perspective.

On the other hand, in some sections there is a lack of critical synthesis of the literature. The text reads more like annotated literature summary. Redundancy is present across sections. Certain citations are repetitive or overused.

I would suggest minor rewording of the abstract for better clarity.

Add a brief mention of the role of the gut-brain axis and psychosocial triggers in Section 2.

I would recommend condensing the macro- and micronutrients subsections.

In section 8-10, reduce historical/cultural background slightly and prioritize nutritional and clinical relevance for IBS. In conclusion, I would recommend including next steps for research, future studies.

Finally, not all the citations are according to the mdpi style.

Author Response

The present paper by Dimitriou et al. addresses a very important problem about the integration of the mediterranean diet into the low FODMAP dietary framework for managing Irritable Bowel Syndrome. The work is comprehensive, well-structured, and is a robust review of the existing literature.

In general, the manuscript is well-written and logically organized, and with strong sectioning. The literature review is up-to-date. The described low FODMAP model is innovative and supported by preliminary clinical evidence. The paper highlights clinical, nutritional, and microbiological considerations, offering a truly multidisciplinary perspective. On the other hand, in some sections there is a lack of critical synthesis of the literature. The text reads more like annotated literature summary. Redundancy is present across sections. Certain citations are repetitive or overused.

Response to Reviewer 1: Thank you very much for your positive feedback on our manuscript. We acknowledge the need for a more critical synthesis of the literature and the reduction of redundancy across sections. We hope that by addressing these points, along with the comments from the other reviewers, we will substantially improve the quality and clarity of the manuscript.

Comment 1: I would suggest minor rewording of the abstract for better clarity.   

Response to comment 1: Thank you for your suggestion. We have revised the abstract to enhance clarity, with a particular focus on more clearly stating the aim of the review. We believe the updated version more clearly communicates the objective and scope of the study.  

Comment 2: Add a brief mention of the role of the gut-brain axis and psychosocial triggers in Section 2.

Response to comment 2: Thank you for the suggestion. A brief mention of the gut-brain axis and psychosocial triggers has been added to Section 2 (2.1, pages 2-3, lines 94-111) to acknowledge their role in IBS pathophysiology and symptom modulation.

Comment 3: I would recommend condensing the macro- and micronutrients subsections.

Response to comment 3: Thank you for the helpful suggestion. The subsections on macro- and micronutrients have been reviewed and condensed to improve clarity and flow, while retaining the essential information relevant to nutritional adequacy in IBS management.

Comment 4: In section 8-10, reduce historical/cultural background slightly and prioritize nutritional and clinical relevance for IBS. In conclusion, I would recommend including next steps for research, future studies

Response to comment 4: Thank you for your valuable feedback. Sections 8–10 have been revised to reduce historical and cultural background and emphasize nutritional and clinical relevance specific to IBS. Additionally, the Conclusion section has been updated to include suggested directions for future research and clinical investigation (page 22, lines 655-657).

Comment 5: Not all the citations are according to the mdpi style.

Response to comment 5: Thank you for pointing this out. All citations have been carefully reviewed and revised to conform to the MDPI referencing style throughout the manuscript.

Round 2

Reviewer 2 Report

Comments and Suggestions for Authors

Thank you very much for following my previous comments.